# Compliance to HIV testing and counseling guidelines at antenatal care clinics in the Kassena-Nankana districts of northern Ghana: A qualitative study

**Wooyon Annette Choi**[1], **Evelyn Sakeah**[2‡], **Abraham Rexford Oduro**[2‡], **John Bosco Achana Aburiya**[3], **Raymond Akawire Aborigo**[2]*

1 Department of Global Health, School of Nursing and Health Studies, Georgetown University, Washington, DC, United States of America, 2 Navrongo Health Research Center, Navrongo, Ghana, 3 Faculty of Integrated Development Studies, University for Development Studies, Wa, Ghana

☯ These authors contributed equally to this work.
‡ ES and ARO also contributed equally to this work.
* rayborigo@yahoo.com

## Abstract

### Background

Utilization of antenatal care services in Ghana has substantially increased over the years, but the rates of mother-to-child transmission of HIV is still high. The high burden of HIV among pregnant women has serious implications for mother-to-child transmission. The main objective of this study was to assess the compliance of HIV testing and counseling provided at antenatal care clinics in two rural districts in northern Ghana by comparing reported practices to the national guidelines.

### Methods

This study was a descriptive qualitative study conducted in the Kassena-Nankana Districts of northern Ghana. In-depth interviews were conducted with 10 midwives, 10 mothers, and 2 public health nurses who were recruited through purposive and snowball sampling. All interviews were audio recorded, transcribed into English, and imported into NVivo 12.0 software for open, axial, and selective coding.

### Results

The findings indicate that not all pregnant women were informed prior to testing nor informed of their test results. Many mothers indicated that pre-test counseling is limited although the midwives claimed to provide it. Post-test counseling is primarily given to those who test positive, and several midwives agreed that there is no need to counsel HIV-negative women. Perceptions of the lack of confidentiality and privacy were pervasive among mothers despite the emphasis placed on its importance by the midwives. There were conflicting reports on whether HIV testing during antenatal care is voluntary or compulsory. The challenges with

**Data Availability Statement:** The interview transcriptions cannot be shared publicly as it contains personally identifiable information and

would therefore undermine the minimal risk ethical committee agreement and terms and conditions of consent. Interviews were confidential to enable freedom of expression, and participants consented to the study with the understanding that only anonymized quotations would be used in publications, not the entirety of the transcripts. Therefore, only illustrative excerpts from the transcripts, which qualify as the minimal data set, are included in the paper. Anonymized data can be made available to qualified researchers by request to the head of the data repository of the Navrongo Health Research Center: Peter Wontou (peter.wontuo@navrongo-hrc.org).

**Funding:** The authors received no specific funding for this work.

**Competing interests:** The authors have declared that no competing interests exist.

HIV testing and counseling that were mentioned by midwives include lack of adequate infrastructure, language barriers, and insufficient training.

## Conclusions

HIV testing and counseling provided at antenatal care is not uniform across all health facilities and does not strictly adhere to national guidelines. Future interventions that focus on standardization, monitoring, privacy, and capacity building are likely to prove valuable in ensuring quality services are provided.

## Introduction

Mother-to-child transmission (MTCT) of HIV accounts for the vast majority of infections in children aged 0–14 years old. This mode of transmission accounts for approximately 16% of all new infections in Ghana [1]. The prevalence of HIV among women aged 15–49 in Ghana is 2.3%, which is twice higher than that of men in the same age group [2]. This higher burden of HIV among women of reproductive age has serious implications for mother-to-child transmission of HIV since a woman infected with HIV can pass on the virus to her child during pregnancy, labor and delivery, and/or breastfeeding.

Ghana has adopted WHO's strategic approach towards prevention of mother-to-child transmission (PMTCT) which focuses on [3]:

1. Prevention of HIV among women of reproductive age

2. Prevention of unintended pregnancies among women living with HIV

3. Prevention of HIV transmission from women living with HIV to their children

4. Provision of appropriate treatment, care, and support to mothers living with HIV and their children

The third and fourth components of this strategy are rooted in Ghana's opt-out policy for routine HIV testing. This approach entails provider-initiated testing and counseling of HIV that is routinely offered to all pregnant women receiving antenatal care (ANC). The test is still voluntary, and women may decline testing services if they so desire.

Previous efforts in Ghana have concentrated more on increasing the utilization of ANC services, rather than the quality of services provided [4]. The proportion of mothers receiving adequate ANC (4–7 ANC visits) increased from 49.3% in 2006 to 58.6% in 2017–2018 [5]. Thus, the proportion of women who use ANC services has substantially increased, but the prevalence of HIV among pregnant women attending ANC clinics is still high at 2.4% nationally in 2016 [6]. The End Term Evaluation of the National HIV & AIDS Strategic Plan for 2011–2015 showed that 70% of all pregnant women were tested and counseled for HIV [1]. However, only 46% of HIV-positive pregnant women received antiretroviral (ARV) prophylaxis to prevent MTCT [7]. Without treatment, the likelihood of MTCT is 15% to 45%, but this probability is significantly reduced to below 5% with antiretroviral therapy (ART) [3]. Studies have found that shortage of drugs, low patient compliance, stigma, drug side effects, and transportation costs may contribute to the low ARV coverage among HIV-positive pregnant women [3, 8]. Research shows that if an HIV-exposed infant is given ART within the first 12 weeks of life, they are 75% less likely to die from an AIDS-related illness [3]. Despite this

knowledge, only 22% of HIV-exposed infants in Ghana received ARV prophylaxis and 18% received Cotrimoxazole prophylaxis [7]. These findings necessitate assessments of quality of HIV testing and counseling (HTC) in Ghana where there is high ANC coverage but high rates of MTCT, leading to high HIV indicators.

In the Upper East Region of Ghana, the median prevalence of HIV among pregnant women has been fluctuating over the years with the highest being 2.4% in 2010 and the lowest at 1.3% in 2017 [6]. Since 2017, the median prevalence has increased to 2.1% in 2019 [6]. In Navrongo, the capital of the Kassena-Nankana districts, the median prevalence of HIV among ANC attendees was 2.0% in 2019, an increase from the last two years [6].

In a study conducted in Bolgatanga, the capital of the Upper East Region, 11.1% of pregnant women attending ANC either didn't think or didn't know if HIV could be transmitted from mother-to-child [9]. Most women (80%) cited breastfeeding as a mode of MTCT of HIV, while 70% of respondents indicated it can occur during delivery [9]. However, far fewer (43%) knew that transmission can also occur during pregnancy [9]. Almost all women (99.3%) were tested for HIV but 8.9% were not counseled on HIV [9]. This may suggest that not all those who were tested were aware of it despite the fact that education and communication campaigns of PMTCT is an integral part of ANC services as outlined in the national guidelines. These findings indicate that there is room for improvement in attaining better PMTCT of HIV indicators.

The main objective of this study was to assess the compliance of HIV testing and counseling guidelines at antenatal care clinics in two rural districts in northern Ghana by comparing reported practices to the national guidelines in order to provide critical information to the health system to improve adherence to the guidelines as well as harmonize practices across the different levels of health facilities.

## Methods

### Study location

Ghana's health care system is organized in four tiers: national, regional, district, and sub-district levels. The community-based health planning and services (CHPS) compounds and health centers at the sub-district level serve as the first point of entry into the health care system. At the district and regional levels, hospitals provide the secondary level of health care. The tertiary level consists of teaching hospitals, which are located in the Northern, Ashanti, Greater Accra, and Central regions of Ghana [10].

HIV-related services include HIV testing and counseling, ART counseling, specimen collection and laboratory testing for CD4, viral load, and early infant diagnosis (EID), and PMTCT services. Ghana's PMTCT program launched in 2002 with two health facilities offering PMTCT services in the Manya Krobo District in the Eastern Region [10]. Since then, Ghana has expanded its PMTCT program to 2748 of over 3765 health facilities providing ANC [10]. There is a total of 197 ART sites in Ghana [10]. In a survey of 172 ART sites, it was found that about 85% percent of health center clinics, 90% of district hospitals and polyclinics, and 100% of tertiary regional hospitals offer PMTCT services [10]. HIV testing and counseling is offered at 96–100% of all levels of facilities [10]. On the other hand, only 6% of facilities conduct viral load testing and 13% provide EID testing because there are only nine viral load machines in the country, which are located in regional/teaching hospitals [10]. Thus, all health facilities in a region share the viral load machine.

The research study was carried out in the Kassena-Nankana West District and the Kassena-Nankana Municipality in the Upper East Region—the third poorest region in Ghana [11]. Subsistence agriculture is the mainstay of the people [11]. The Navrongo Health and Demographic Surveillance System (NHDSS), which is operated by the Navrongo Health Research Center

(NHRC), has divided these districts into five zones: North, South, East, West, and Central. There are two main ethnic groups in the districts—the Kassenas and the Nankanis. The Kassenas mainly inhabit the North and West zones while the Nankanis predominantly live in the East and South zones. The Municipality and West district each have one district hospital and share eleven health centers, over forty community-based health planning and services (CHPS) compounds, and one private clinic.

## Ghana national guidelines on PMTCT

Routine HIV testing and counseling during pregnancy is an essential component of HIV prevention, treatment, and care programs in many sub-Saharan African countries because it is a critical entry point for PMTCT of HIV [12]. Ghana's Ministry of Health published national guidelines for PMTCT in 2014, which outlines the approach for the provision of PMTCT services. According to these guidelines, all pregnant women are supposed to be offered HTC services at registration. However, they have the right to refuse testing. When a woman declines testing, the midwife is expected to routinely offer HTC services during subsequent visits. On the other hand, if she consents to testing, the First Response test is conducted to determine the initial diagnosis. The midwife is then supposed to inform the client of the result. If the test result is negative, the woman is to be further counseled on how to remain uninfected and the test should be repeated in the third trimester to confirm the final diagnosis. If the test result is positive, the guidelines provide for a confirmatory test using the Oral Quick diagnostic test. If the Oral Quick also indicates an HIV-positive status, the client is to be immediately initiated on ARVs, receive post-test counseling, referred to a higher-level facility to conduct further immunological assessments, and receive follow-up care. For HIV-exposed infants, the national guidelines instruct midwives to initiate ARVs within 48 hours. The dried blood sampling should be taken at 6 weeks to determine the initial HIV-status, and serological testing should be repeated at 18 months to confirm the final diagnosis (Fig 1). Couple and partner HTC, including disclosure, should be encouraged and offered regardless of the HIV status of the woman.

## Study design

This was a descriptive qualitative study based on the Straussian grounded theory approach. Grounded theory focuses on social relationships, social processes and actions, and understanding the underlying processes of what is going on. We used in-depth interviews (IDIs) with health workers and caregivers to elicit information on the processes of PMTCT at ANC, perceptions of the current processes, and experiences of HIV testing and counseling at ANC. We utilized the fundamental characteristics of the Straussian grounded theory approach, namely theoretical sampling, thematic saturation, constant comparative analysis, and memoing to guide the sampling approach, data collection, and data analysis [13].

## Study population

Target population: Based on our knowledge of the cadre of health workers who provide ANC, our initial target sample was midwives and the mothers who use their services. Through theoretical sampling, we added public health nurses and members of the district health management team as target populations.

## Sampling approach

From the demarcations made by the NHDSS, two zones, each representing the Kasem or Nankani ethnic group, were randomly selected in order to document ethnic variations if any. The

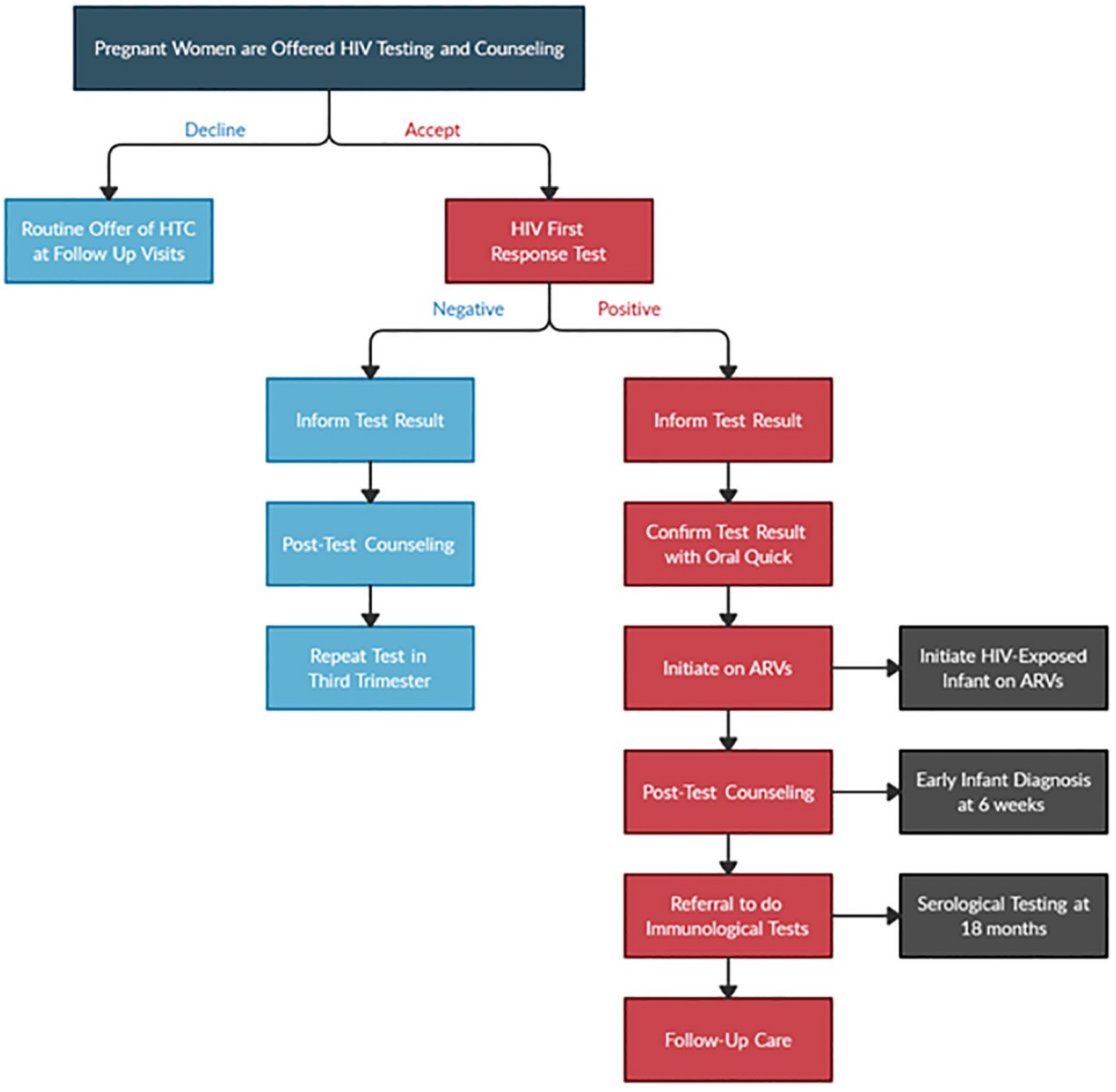

**Fig 1. Flowchart depicting Ghana national guidelines on PMTCT.**

South zone was randomly selected to represent the Nankanis while the West zone was selected for the Kassenas.

In each selected zone, a list of all health facilities was prepared. We purposively selected five health facilities with a midwife providing antenatal care from each zone—. three CHPS and two health centers. The names and contact information of all midwives working in the selected health facilities were obtained from their respective DHMTs. We visited and interviewed the midwives until we reached thematic saturation.

Through the theoretical sampling approach, we identified the public health nurses as a relevant source of information to answer our research question. Each of the DHMTs has one public health nurse and both of them were interviewed.

Through the snowball sampling approach, the midwives helped the research team identify recently delivered (6 weeks postpartum) women who used their services. This time restriction was to enable mothers recall details of their HTC experience during pregnancy.

## Data collection

Two interview guides were designed for the study–one for health workers and one for caregivers. The content of the tools was determined by the research objectives. We pre-tested the guides at War Memorial Hospital in Navrongo with one mother and one midwife to assess the appropriateness of the questions and to determine the approximate time required to complete an interview. We revised the guides based on the interviewers' experiences and observations. In addition, new questions were added to the guides upon discovery of new themes during the data collection phase.

One research assistant (JA) conducted the interviews with recently delivered mothers in their preferred languages—either Kasem, Nankam, or English—while the lead investigator (WC) conducted the interviews with the midwives and public health nurses in English. Both researchers used the English interview guide, but the research assistant (JA) who conducted interviews with the mothers verbally conveyed the questions in the local language. Both interviewers had experience conducting IDIs and therefore were trained for five days to hone their skills. Training included both didactic sessions and mock interviews with real-time feedback to ensure neutrality in interviewing and the ability to probe effectively for additional information. The training included an explanation of the study, the consenting procedures, issues around confidentiality, interviewing, and probing. The research assistant (JA) was extensively trained by an experienced field interviewer to accurately translate the questions, identifying key terms that may not have a direct translation to harmonize data collection between interviews.

We conducted the interviews after explaining the purpose of the research, the procedures involved, the voluntariness of participation, the risks and benefits of the research, and the right to withdraw from the study to potential participants and obtaining their consent. We also sought consent to audio record the interviews. We conducted the interviews with mothers in their homes in a private setting in the absence of other family members, and the interviews with midwives at the health facilities. We scheduled interviews with midwives at their convenience in order to not disrupt service provision. In all, we conducted 22 IDIs; 10 with recently delivered mothers, 10 with midwives, and 2 with public health nurses at the DHMTs. The interviews were conducted from October 31, 2019 to November 19, 2019.

We used the constant comparative approach to monitor thematic saturation, which is defined as the point at which the data collection process no longer offers any new or relevant data [14]. This ensured that the data collected was exhaustive and trustworthy. Based on the literature, scholars have suggested anywhere from 6 to 50 participants for in-depth interviews to be sufficient to reach thematic saturation [15–17]. In the current study, the researchers constantly compared information from new interviews to previous interviews to identify new themes or new categories of participants who could provide different perspectives to the research question.

## Data processing and analysis

All interviews were transcribed verbatim into English and edited to ensure flow by an experienced transcriber. The transcripts were reviewed for accuracy and completeness and corrected to facilitate coding by theme. The transcripts were then imported into QSR Nvivo 12.0 software for open, axial, and selective coding. Multiple readings of the transcripts were conducted

in order to gain a better understanding of the ideas and concepts arising from the texts that were written down in the form of memos, which allowed us to identify themes that were recurrent across interviews as well as those that were not frequent. The main codes were first identified using the questions on the interview guide, and the memos were used to develop coding categories. Descriptive coding was subsequently used to create a codebook, and simultaneous coding was applied due to multiple topics being present in the same text. Additional codes and subcodes were added to the codebook as coding progressed. Using grounded theory, thematic analysis was conducted, and the themes were used to generate a conceptual model to understand the process of HIV testing and counseling.

## Statement of ethics

We obtained ethics approval from the Navrongo Health Research Center Institutional Review Board (NHRCIRB358). We also received permission from the Directors of the District Health Management Teams (DHMTs) and the heads of the health facilities to interview the health workers. The NHRC Institutional Review Board assessed risks associated with participation in the study to be minimal and therefore we used oral consent to recruit participants into the study.

# Results

## Characteristics of respondents

Table 1 gives a summary of the characteristics of the respondents.

## Process of HIV testing and counseling at ANC

All recently delivered women claimed that they were tested for HIV during their ANC visits. While some midwives usually inform their clients prior to testing, others do not. The midwives reported a number of required tests during ANC which makes it difficult for them to inform each client of every test that is done. Consequently, tests are written in the pregnant woman's ANC book without verbally informing her about the tests conducted and why. The dialogue below confirms this assertion.

> "*I*: *Did they test you to know whether you have the HIV disease or not when you were attending ANC?*
>
> *R*: *They tested, it seems they tested.*
>
> *I*: *How do you know they tested*? *Because you are not sure they tested, you are doubting.*
>
> *R*: *You know they always write in the folder for you and I always look, but I don't understand their writing very well. So, it is from the writing that I got to know that they tested.*"
>
> **(Mother 04)**

Other reasons for not pre-informing clients was to avoid scaring them and creating an opportunity for clients to decline the test. Based on the client's reaction during pre-test counseling, midwives determine whether to inform them about the test or not. There were reports that due to fear of being stigmatized, some clients who are aware of their HIV status refuse to disclose to family members, including health workers.

> "*When you are giving the pre-information, the person's responses will influence me telling her whether I'm coming to do the test. Because I realized, some of them are aware but they*

**Table 1. Demographic characteristics of respondents.**

| Characteristic | Mothers (N = 10) | Midwives (N = 10) | Public Health Nurses (N = 2) |
|---|---|---|---|
| **Age, years, mean (range)** | 25 (16–35) | 38 (28–55) | 45 (33–57) |
| **Gender, N (%)** | | | |
| Female | 10 (100) | 10 (100) | 1 (50) |
| **Educational Level, N (%)** | | | |
| Primary | 3 (30) | - | - |
| Junior Secondary School | 5 (50) | - | - |
| Senior Secondary School | 1 (10) | - | - |
| Tertiary | 1 (10) | - | 2 (100) |
| Midwifery | - | 10 (100) | - |
| Trained on HTC | - | 6 (60) | 1 (50) |
| **Parity, N (%)** | | | |
| 1 | 1 (10) | - | - |
| 2 | 4 (40) | - | - |
| 3 | 1 (10) | - | - |
| 4 | 4 (10) | - | - |
| **Years Worked as a Midwife, median (range)** | - | 4.5 (2–26) | - |
| **Years Worked as a Public Health Nurse, median (range)** | - | - | 10.5 (1–20) |

*wouldn't want you to know her status. Even during the counseling, she will still not want to do the testing though she is aware. And she will not openly tell you that, 'oh madam this is my status,' so if I should read and I realize that she's that kind of person or I am suspecting so, then I don't inform her. I do my testing just for my personal use."*

**(Midwife 02)**

Clients who decline the HIV test are provided further counseling and referred to the nearest ART center. Women who refuse are captured in the ANC records. One midwife indicated that she would occasionally bring in an example of an HIV-positive woman to show to her clients who are hesitant about getting tested. All other midwives did not approve of this approach and cited reasons of stigma. Some midwives reported if that is necessary, the expressed permission of the HIV-positive woman must be sought before such exposure. Here is what the midwife who uses a sero-positive pregnant woman to encourage other women to go through the test had to say.

*"If a client is resisting to go for the test, I just invite an HIV-positive woman. Then she will come. 'Oh, she is having it and she is on the drug. Even if you do and you are positive, they will put you on the drug and as you are taking the drug, you will look healthy.' So they should look at her, she is fine and all that. So you see that from there, they will feel happy and they will allow themselves to be tested."*

**(Midwife 08)**

Most midwives disclose HIV test results to their clients. The few who fail to do so said it was because they do not inform the clients about the test. Prior to testing, the midwives usually explain to the clients the meaning of the lines in the test kits. The clients are then shown the cassette used for the test to interpret the result. The midwives usually confirm the interpretation before proceeding to provide post-test counseling. Here is what one of the midwives had to say.

*"When we do it, we explain it to you. When you see one line at the control, it means it's negative and when you see two, one at the testing and one at the control, it's positive. So when the test is done and the results are ready, you will show it to the client to tell you what she has seen on it. She will tell you whatever lines she has seen, and you will ask her the number of lines and what she has seen. So the results will come out from what she has seen on the kit. I'm not going to tell her that she is positive or negative and throw the test kit away."*

**(Midwife 06)**

Mothers who are not shown the cassette to interpret the results for themselves are mostly those whose tests are not conducted by the midwife. Such women are referred to laboratories for their tests, and the professionals in those laboratories do not give them such access.

The midwives claimed that they usually repeat negative tests at least once between 34–36 weeks, while some repeat it twice: one at 28 weeks and the other at 36 weeks. They said the test is repeated because some of the women may be in the early stages of infection while some may have low viral loads, which may escape detection using the First Response test. The excerpt below illustrates this.

*"So when we get you and then counsel and test you for the first time and you are negative, when you get to 34 weeks, I'm supposed to do another test. Probably maybe the time that you tested, the HIV has not gone through your whole system. So, it may be possible that you have it, but it has not showed. So, probably when you get to 34 weeks and then we do the testing again, it can be that the results can be positive."*

**(Midwife 09)**

Routine HIV testing at ANC is done using the First Response test kits. The midwives reported that apart from HIV, other things could trigger a positive result from using the First Response test kit. In view of that, the confirmatory tests are done using the Oral Quick, which is more reliable. A positive result from the Oral Quick triggers initiation of treatment using ARTs. ARVs are readily available at the district hospital and select health centers. They are not stored at the CHPS compounds because they may expire and go to waste given the lower volume of HIV-positive clients. The midwives working at the CHPS compounds indicated that in the event that a pregnant woman tests positive for HIV, they will obtain the ARVs from the nearest ART center to give to their client. For the health centers with an ART clinic, positive clients are referred there for continuity of care. Midwives noted that the drugs are always available, and shortages are extremely rare.

*"R: As for the ARVs, when we started, we were given some to keep at our ANC and administer but they (the DHMT) realized that most people do not get the cases. So, the drugs stay there, instead of returning them when they are almost expired. They let them expire and they throw them away. So, they (the DHMT) are now saying, if we get the case, we should come for the drugs every month for the woman."*

**(Midwife 07)**

*"I: Are the ARVs always available when you go and take it?*

*R: Yes, there hasn't been a time that it has run out."*

**(Midwife 02)**

In addition, the midwife tells the client with a reactive result to invite her partner for testing to prevent transmission within the family. The partners of HIV-negative women are typically not invited for testing and counseling. However, some midwives mentioned that regardless of the HIV test result, they still invite the partner because even if they are negative, they can acquire the infection at any time, and counseling can better equip them to protect themselves. They noted that this was particularly important in the research setting where polygamy is common. Here is what one of the public health nurses had to say.

> *"Usually if the man is infected and you don't counsel them well, they go about just living their I don't 'careism' life. I mean that is jargon. They live their life freely. At least, for a man, he doesn't have to be with one woman, so they will continue having many partners and they infect others and they re-infect themselves. So, it is good to counsel the partner to also know his status and then let him know how to prolong his life."*

**(Public Health Nurse 01)**

Although not a common occurrence, there are times when women come from other health facilities to deliver or when a woman comes in during labor without having the HIV test done. If the delivery is not imminent, midwives will conduct the HIV test. However, if the woman is in the second stage of labor, they will conduct the delivery first and do the test subsequently.

> *"I earlier said that others will start their ANC somewhere and they didn't test, meaning when she comes in second stage and I deliver her, she still doesn't know her status. So, it's after the delivery that I will test because maybe the stage in which she came, the child is also coming. So, I can't say, 'oh child wait and I will do my HIV test.' So, some will deliver before they know only if they were not tested if they were not weighing here and they were not tested where they weighed, but they have come in labor. But if it is early in labor, we test before she delivers."*

**(Midwife 03)**

Another uncommon but still existing situation is the case of an inconclusive test result. Typically, the midwives will try confirming the result with the Oral Quick but when that fails, they refer to the next level of care.

> *"Just yesterday I had to do it three times, on three occasions I did it for the client. When she came at registration, it was negative, but at 34 weeks she came and then there was a faint line on the type two. So, I did the Oral Quick and it was negative. Then I had to do the first response again. And there was still the thin line, so I was like what will I do. So I had to call the senior midwife at War Memorial Hospital to ask her what to do. And she said I should refer her there and that they will take it from there. So I just referred her."*

**(Midwife 07)**

## Information provided during HIV counseling

During the pre-test counseling, some midwives first test the knowledge of the client on modes of transmission and preventive measures for HIV. They usually supplement the knowledge of the client during counseling by sharing information with them on the importance of testing, procedures for screening, the consequences of a positive result for the mother and baby, and availability of treatments. Some mothers indicated the midwives did provide this information before testing, while others said they did not.

*"Mostly for new clients or registrants when we give the card, we ask whether she knows her HIV status. So, whether she knows or she doesn't know, we will tell her we need to test so that we will be able to prevent her from transmitting the HIV to the child. So, we ask whether she knows anything concerning HIV, how it is transmitted, what she will do to prevent herself from the infection. So, mostly, depending on the feedback they give us, we now educate them on how the disease is acquired."*

**(Midwife 10)**

*"They advised me to keep myself away. Through sex I can also be affected by HIV or through blade and blood. Somebody may be wounded and I have to help, though at the village you won't get gloves. But what you will do is you use rubber to tie your hands before you help the person so that you will not be infected. Because you don't know whether you have a wound or not, so that was the advice they gave to me."*

**(Mother 03)**

The midwives said they usually educate their clients on mother-to-child transmission as it occurs during pregnancy, labor, delivery, and breastfeeding. As part of the education, they explain to the mothers that blood exchange between the mother and child occurs through the placenta. Thus, if the client is HIV-positive and is on ARVs, the viral load is reduced so there is a lesser chance of transmitting it to the infant. Also, during labor and delivery, *"if instruments are not well kept or the membranes rupture and then if the separation of the cord is not done well, the child can get it."* **(Midwife 03)**

They also explained how HIV transmission occurs during breastfeeding.

*"I tell them that when the teeth start coming, there is a chance of the child biting the nipple and there will be blood and the child can suck the blood and be infected through that. So, mostly we discourage breastfeeding after one year when the child starts developing teeth."*

**(Midwife 07)**

The mothers confirmed the information given to them by the midwives in the excerpts below:

*"I: Were you told that you can also transmit the virus to the child during labor and delivery?*

*R: Yes, that one too my child can also be affected through delivery, especially if I delivered and they are going to cut the cord. Through that one, if they didn't tie it well, my child can also be affected. Or through the blood depending on how they handle the child before me."*

**(Mother 03)**

*"They said it, through childbirth or breastfeeding. Even if I deliver and the child is breastfeeding, they have a timeline which they will not allow the child to breastfeed again. If I don't abide by that timeline and decide to breastfeed the child way longer than the time they have given, it can make the child acquire the disease. They know the disease and they know how it is, and they gave me a timeline not to breastfeed the child."*

**(Mother 05)**

The midwives explained that knowing the woman's HIV status during ANC guides the kind of attention that she receives whenever she visits the health facility. Although most of the mothers understood these reasons, some were unsure as to why they were tested for HIV during ANC as contained in the quote below:

*"I don't know the reason, but I think they do it so that if you are infected, they will prevent it from infecting the baby. That is why they do it."*

**(Mother 05)**

Mothers reported mixed messages on breastfeeding by HIV seropositive mothers. One mother reported the mixed messages as follows:

*"You are not to breastfeed the child, but I learnt some say that the child will breastfeed for six months and then stop breastfeeding. And others are saying that, you don't breastfeed at all because if the child breastfeeds s/he will also get the disease."*

**(Mother 08)**

Some mothers claimed not to have received any message on breastfeeding by HIV-positive mothers.

HIV-negative women receive counseling on preventive measures, such as using condoms and being faithful to their partners. In addition, the midwives tell them the test will be repeated at nine months to confirm their HIV-status before delivery.

For HIV-positive women, the midwives said they counsel them on safe infant feeding practices, drug adherence, and how to prevent transmission within the home. They also encourage them to deliver in a health facility in order to reduce the risk of mother-to-child transmission. In addition to these prescriptive behaviors, the midwives said they provide the HIV-positive mothers with some psychological support. Some midwives said they provide post-test counseling regardless of the HIV status of the client, but some do not as contained in the excerpt below:

*"Actually, the counseling we don't go into details like that. So, that one let me be frank with you. After explaining to them and doing the test and the person is negative, I don't go further. I only emphasize on those who are positive."*

**(Midwife 10)**

## Lack of privacy and breach of confidentiality

Confidentiality was heavily emphasized by the midwives. They said the whole HIV testing and counseling process is one-on-one and that results are not disclosed to a third party, unless the client gives them permission to do so. While some mothers indicated that the disclosure of their HIV test results were done privately, others reported breaches in confidentiality as noted in the excerpt below.

*"I: When she disclosed your status to you, was it between the two of you or were there other people around when she disclosed your results to you?*

*R: Her colleagues were seated. I was sitting below the table and they were seated around the table and she told me. When she started to do the test, they were around. So when she brought*

*back the results, they were still seated and she disclosed to me that I don't have it. She said it in the midst of other people. I wasn't the only one."*

**(Mother 02)**

## Voluntariness of HIV testing

There were conflicting reports on whether HIV testing during ANC is voluntary or compulsory. A few midwives reported that clients have the right to refuse testing after receiving counseling on HIV and PMTCT. However, most midwives said the test is compulsory for all pregnant women because they want to eliminate the risk of mother-to-child transmission and to take the necessary precautions to protect themselves. There were reports that when clients refuse, some midwives still conduct the test. The reason for this is captured in the quote below:

*"She doesn't have a choice because at the end of it, I'm involved. So if you denied me of testing and assuming you are positive and I should get a needle prick and you are denying me of testing to know your status, then you put me at a higher risk or I will panick more. But if I know your status and then I get a needle prick, I will not panic so much, so it's good to do the testing."*

**(Midwife 02)**

The conflicting views on the voluntariness of HIV testing during ANC was confirmed by the mothers as follows:

*"They said if you don't wish to be tested, they can't test you but if you want to know your status, then they can do it for you."*

**(Mother 06)**

*"They will just say we want to test you. It's compulsory. You cannot say no. If you are a pregnant woman, they will test you to see whether you are negative or positive."*

**(Mother 07)**

## Challenges with HIV testing and counseling

Several midwives stated the lack of adequate infrastructure to maintain confidentiality hindered them from comprehensively counseling the client. Many health facilities, especially the lower level CHPS compounds, do not have a room dedicated to ANC clients. Additionally, the infrastructure is not soundproof, so people nearby can easily eavesdrop interactions between the nurses and their clients.

*"To me, confidentiality is key. You get to a facility and there is no place to sit and counsel the client. Whatever you say inside everybody hears it. So to me, if all the facilities where PMTCT is done is provided with structures that give the client and the midwife some confidentiality, I think that will improve the testing. Because most at times, some will not like to give much information because they think some people sitting outside might hear what they are saying. But if there is a place built purposely for that, you go there with the client alone, the client is free and relaxed."*

**(Midwife 07)**

Some midwives indicated they have difficulties communicating with clients due to language barriers. There are two predominant local languages spoken in the Kassena-Nankana districts. However, some midwives are not from the districts or only speak one of the two languages.

*"The challenges that I have been facing is the language barrier. Sometimes you get a client, you want to express yourself, but the person cannot speak English, cannot speak Twi, and is a Kasem. Me too, I cannot speak Kasem and you know this is a sensitive thing and you cannot call in a third party to come in. So sometimes you always find it difficult. Let's say if the person is positive and you want to explain to the person, you call another person to come, but the person will not be comfortable."*

**(Midwife 04)**

Most of the midwives reported receiving training on HIV testing and counseling. Those who have not been formally trained received informal training from a senior midwife or from the HIV coordinator at War Memorial Hospital. While those who have been trained indicated that they generally feel qualified to provide HTC services, the untrained said they feel less qualified as exemplified by this quote:

*"I think that I am not well equipped to be doing HIV counseling and testing. It's just the knowledge that I have that I use to do my counseling and testing."*

**(Midwife 10)**

Despite efforts to train midwives on HTC, the general consensus about the length of the training was that it was too short. The training sessions usually last a week which was reported to be inadequate for trainees to assimilate all the gamut of information provided. Midwives complained that the course is too loaded and intensive with the focus going beyond HIV.

*"Our time, we were thinking it's too short because in a day, so many topics will be treated. So, if you are not sharp, it's very hard to get everything. So, we were saying if it were to be two weeks, then we will take time and absorb the topics bit by bit. After the course, we did an evaluation and said the time was too short for the course because it was so loaded. So, if they were to use two weeks it would have also benefited us."*

**(Midwife 07)**

The training is usually organized at the regional level with priority given to providers who have never been trained.

## Discussion

This study aimed to describe the process of HIV testing and counseling at ANC in Ghana in order to determine if the provision of care is meeting the national guidelines. Overall, the general flow of the process of HIV testing and counseling loosely adheres to the national guidelines. However, there are many discrepancies in implementation across different health facilities. The findings indicate that not all pregnant women were informed prior to testing nor informed of their test result. Many mothers indicated that pre-test counseling is limited although the midwives claimed to provide it. Post-test counseling is primarily given to those who test positive, and several midwives agreed that there is no need to counsel HIV-negative women. Perceptions of the lack of confidentiality and privacy were pervasive among mothers

despite the emphasis placed on its importance by the midwives. There were also conflicting reports on whether HIV testing during antenatal care is voluntary or compulsory.

According to the current national guidelines, women must be explicitly informed of their right to refuse testing, and their decision to decline testing services must be respected [12]. However, some women do not perceive HIV testing as a choice but rather as a compulsory service. The perception that HIV testing as part of ANC is compulsory has also been documented in other regions of Ghana, such as Kumasi, as well as other African settings, such as Uganda, Ethiopia, and Malawi [18–21]. Similarly, in a study conducted in Ethiopia, health providers were observed taking blood samples for HIV testing from clients without informing them [22]. This is problematic from an ethical and human rights perspective because neglecting informed consent and disregarding patients' autonomy creates imbalanced power dynamics between the health worker and patient.

The lack of confidentiality and privacy is pervasive at the lower-level health facilities due to the poorly designed layout of the infrastructure and how disclosure of HIV test results is done in the presence of other people. This presents several challenges that compromises the quality of care pregnant women receive. For example, the lack of confidentiality may deter women from utilizing ANC services, exacerbate their reluctance to accept HIV testing, hinder health workers from providing comprehensive HIV counseling due to fear of being heard by others, and increase clients' unwillingness to share personal health information. Similar concerns were raised in another study in southern Ghana which found that respondents who were unsure about the maintenance of privacy at the time of counseling were five times more likely to refuse HIV testing as compared to those who believed privacy was maintained at the time of counseling [18]. Patient confidentiality is one of the most important pillars of healthcare as it impacts the level of trust between the health provider and patient as well as encourages patients to be open and honest about sensitive information. Additionally, in a setting where stigmatization of people living with HIV is still predominant, it becomes more critical to ensure confidentiality to minimize the negative effects of stigma that can ensue.

Effective counseling on HIV is critical in guiding decisions and in increasing awareness. According to the national guidelines, clients must receive pre-test counseling to ensure they understand the purpose and benefits of testing as well as post-test counseling regardless of HIV status. In our study, however, many caregivers indicated that pre-test counseling is limited although the midwives claimed to provide it. It is evident from the results of this study that inadequate pre-test counseling has contributed to varying levels of accurate knowledge among pregnant women, leaving them uncertain about proper conduct and awareness of related health issues. Similar to findings from a study in Ethiopia, the midwives acknowledged that post-test counseling is necessary for HIV-positive women [20]. They also agreed that there is no need to counsel HIV-negative women since the goal of HIV testing is to prevent mother-to-child transmission [20]. This implies that no counseling is provided to pregnant women who test negative. Considering that more women test negative, midwives miss the opportunity for primary prevention of HIV. While it is undeniably important that HIV-positive women need special attention and care to reduce the risk of MTCT, it is equally imperative for HIV-negative women to receive post-test counseling on how to remain uninfected because a negative result does not mean immunity to HIV.

The decision to invite the partners of only pregnant women who are sero-positive for HIV counseling further limits the opportunity of curtailing the spread of the disease in the family. The fact that the woman tested negative is no indication that the partner is also negative and therefore both the woman and the child may not be wholly protected throughout pregnancy and the postpartum period. This is particularly important in patriarchal settings such as the

study site, where polygamy is common, thus increasing the risk of infection [23]. As noted earlier, this approach also misses the opportunity for primary prevention within the home.

Prevention of mother-to-child transmission of HIV guidelines and best practices are constantly being revised as more evidence-based research emerges. For example, WHO has revised PMTCT guidelines over five times since 2001 [24]. Thus, midwives need to be kept up to date with the latest knowledge through continuous training throughout their service. However, this study highlights that this is a challenge in Ghana where some respondents were not formally trained on HIV testing and counseling and opportunities to attend in-service training are not equally distributed due to financial constraints and limited resources. Although midwives do undergo training, challenges with the current training approach leaves some of them feeling incompetent to provide HTC services. Midwives may need refresher courses to update their knowledge and remind them of the essentials in providing the services. Our findings are in accordance with a study done in Malawi, which also found that 70% (n = 57) of nurses were trained in PMTCT, but unlike this study, 74% of those trained had received in-service training [25]. On the other hand, it was found that only 43% of participants in a Nigerian study had been formally trained [26]. Insufficient training on HIV testing and counseling has far-reaching effects beyond simply the competency of health workers to convey accurate information and confidently provide appropriate care to HIV-positive women. When considering the patients' perspectives, perceptions of uneducated and unqualified health workers may deter clients from subsequently utilizing health services and may result in low patient satisfaction, which can consequently impact compliance and trust in health workers.

## Strengths and limitations

It is important to understand how routine HTC is perceived and experienced to ensure that the HIV opt-out policy is implemented in a way that maintains quality. The use of in-depth interviews in this qualitative study enabled us to gain a detailed understanding of compliance to national PMTCT guidelines at ANC and both mothers' and health workers' perceptions and experiences of HTC. Conducting interviews with both health providers and recipients of health services allowed us to triangulate the findings to determine if there were any commonalities and/or discrepancies, which helps further elucidate the reality of the situation.

The findings of this study should be interpreted in the light of the following potential limitations. Social desirability may have influenced some of the responses from study participants. Reporting bias among the health workers may have impacted the trustworthiness of their responses because what they claim may not directly translate into what they practice. Thus, a study design like ethnography or observational studies may better capture the reality of HTC. In addition, the mothers may have been influenced by recall bias. Although the eligibility criteria was set to enable only recently delivered mothers with newborn infants not older than 6 weeks to be sampled to maximize recall challenges related to their HIV testing and counseling experiences, there were still a few mothers who responded that they could not remember for some questions. The mothers were recruited using snowball sampling, which may have also introduced bias because the midwives may have suggested those whom they think can speak best about their HTC experience. This study was conducted at lower-level health facilities, primarily CHPS compounds and health centers, with fewer staff and poorer facilities. Thus, the inconsistencies and gaps identified in this paper may not be as pronounced at higher-level health facilities, such as the district and regional hospitals.

## Conclusion

HIV testing and counseling provided at antenatal care is not uniform across all health facilities and does not strictly adhere to national guidelines. Experience with HIV testing and counseling differed between facilities with consistent reports among mothers and midwives. Pre-test counseling is currently poor as it fails to ensure that HIV testing is implemented in a way that preserves clients' autonomy and maximizes opportunities for primary prevention. These findings call for standardization of HTC to ensure all health workers are following the same protocols. An effective monitoring mechanism should be implemented in order to guarantee compliance to the protocols and ensure quality services are provided by routinely incorporating feedback from clients. Training on HTC should be standardized and provided to all midwives so that they all receive the same information before being deployed to their respective health facilities. Strengthening capacities at the district level to provide such training can ensure more midwives are up to date. In order to reduce language barriers, midwives should be deployed to health facilities based on their language abilities. Finally, future design of ANC infrastructure should consider the privacy of the client in order to protect confidentiality. Policy makers and health advocates should reflect upon the findings of this study as evidence of a p7ressing need to improve HIV testing and counseling services at ANC and consider these recommendations as possible interventions and strategies. Further research, such as ethnography and observational studies, should be conducted to describe actual practices of HIV testing and counseling services provided at ANC clinics.

## Supporting information

**S1 File.**
(DOCX)

**S2 File.**
(DOCX)

## Acknowledgments

The authors would like to recognize the support of Navrongo Health Research Center and the School of Nursing and Health Studies at Georgetown University. Our appreciation also extends to the District Health Management Teams and the people of the Kassena-Nankana District whose support and cooperation enabled us to successfully carry out the study.

## Author Contributions

**Conceptualization:** Wooyon Annette Choi, John Bosco Achana Aburiya, Raymond Akawire Aborigo.

**Data curation:** Wooyon Annette Choi, John Bosco Achana Aburiya, Raymond Akawire Aborigo.

**Formal analysis:** Wooyon Annette Choi.

**Investigation:** Wooyon Annette Choi, Abraham Rexford Oduro, Raymond Akawire Aborigo.

**Methodology:** Wooyon Annette Choi, Evelyn Sakeah, John Bosco Achana Aburiya, Raymond Akawire Aborigo.

**Project administration:** Wooyon Annette Choi, John Bosco Achana Aburiya.

**Resources:** Abraham Rexford Oduro.

**Supervision:** Evelyn Sakeah, Abraham Rexford Oduro, Raymond Akawire Aborigo.

**Validation:** Wooyon Annette Choi, Evelyn Sakeah, John Bosco Achana Aburiya, Raymond Akawire Aborigo.

**Writing – original draft:** Wooyon Annette Choi.

**Writing – review & editing:** Evelyn Sakeah, Abraham Rexford Oduro, John Bosco Achana Aburiya, Raymond Akawire Aborigo.

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
