## [Decision Letter · Decision Letter 0]

4 Feb 2021

PONE-D-20-35332

Compliance to HIV testing and counseling guidelines at antenatal care clinics in the Kassena-Nankana districts of northern Ghana: A qualitative study

PLOS ONE

Dear Dr. Aborigo,

Thank you for submitting your manuscript to PLOS ONE. After careful consideration, we feel that it has merit but does not fully meet PLOS ONE’s publication criteria as it currently stands. Therefore, we invite you to submit a revised version of the manuscript that addresses the points raised during the review process.

We look forward to receiving your revised manuscript.

Kind regards,

Juliet Kiguli, MA, PhD

Academic Editor

PLOS ONE

Additional Editor Comments (if provided):

Please address all comments carefully and resubmit.

Journal Requirements:

2) Please include additional information regarding the qualitative questionnaire used in the study and ensure that you have provided sufficient details that others could replicate the analyses. For instance, if you developed a questionnaire as part of this study and it is not under a copyright more restrictive than CC-BY, please include a copy, in both the original language and English, as Supporting Information, or include a citation if it has been published previously.

3) In the Methods, please discuss whether and how the questionnaire was validated and/or pre-tested. If these did not occur, please provide the rationale for not doing so.

4) We note that you have indicated that data from this study are available upon request. PLOS only allows **data to be available upon request if there are legal or ethical restrictions on sharing data publicly. For** information on unacceptable data access restrictions, please see http://journals.plos.org/plosone/s/data-availability#loc-unacceptable-data-access-restrictions.

Reviewers' comments:

Reviewer's Responses to Questions

**Comments to the Author**

1. Is the manuscript technically sound, and do the data support the conclusions?

Reviewer #1: Yes

2. Has the statistical analysis been performed appropriately and rigorously? 

Reviewer #1: N/A

3. Have the authors made all data underlying the findings in their manuscript fully available?

Reviewer #1: Yes

4. Is the manuscript presented in an intelligible fashion and written in standard English?

Reviewer #1: Yes

5. Review Comments to the Author

Reviewer #1: 1. Introduction section: Update literature on prevalence of HIV among pregnant women at ANC in study area . There is also relevant literature in Ghana e.g by Osei et al 2015 on prevention of MTCT of HIV and should be included. Probable reason for very low uptake of ARV services among HIV-positive pregnant women should be stated.

2. Study location: Provide more information on the health system context by way of facilities or services available at the different levels of health facilities especially as it relates to ANC and prevention of MTCT. Dates or period study was conducted should be stated.

3. Population and sampling approach: Consider systematically organizing this section into paragraphs of first the study population, sampling at the district level, zones and health facilities by levels of care for clarity. Please indicate the reasons for the sampling approaches used for the different populations.

4. Results: Please provide demographic characteristics of study participants. Please include information on availability of treatment for the prevention of MTCT of HIV since it could also impact on compliance to the guidelines being assessed. Additionally it could be an important discussion point.

5. Consenting: Reason provided for oral consenting is not accurate since most respondents were health workers and may be considered as a limitation instead.

6. PLOS authors have the option to publish the peer review history of their article (what does this mean?). If published, this will include your full peer review and any attached files.

Reviewer #1: **Yes: **Livesy Naafoe Abokyi (PhD)

---

## [Author Response · Author response to Decision Letter 0]

6 Mar 2021

Comments from the Editor:

https://journals.plos.org/plosone/s/file?id=wjVg/PLOSOne_formatting_sample_main_body.pdf
https://journals.plos.org/plosone/s/file?id=ba62/PLOSOne_formatting_sample_title_authors_affiliations.pdf

Response: We have reviewed both PDFs detailing PLOS ONE’s style requirements and have edited the manuscript to reflect this. 

2) Please include additional information regarding the qualitative questionnaire used in the study and ensure that you have provided sufficient details that others could replicate the analyses. For instance, if you developed a questionnaire as part of this study and it is not under a copyright more restrictive than CC-BY, please include a copy, in both the original language and English, as Supporting Information, or include a citation if it has been published previously.

Response: We have uploaded a copy of the qualitative interview guide for mothers and health workers in English as Supporting Information. During the data collection process, the research assistant (JA) who conducted interviews with the mothers in their preferred language used the English interview guides but verbally conveyed the questions in the local language. This research assistant was extensively trained by an experienced field interviewer to accurately translate the questions from English to the local languages, identifying key words in English that may not have a direct translation. This training included mock interviews in the local language with real-time feedback to ensure understandability and veracity of the translation.

3) In the Methods, please discuss whether and how the questionnaire was validated and/or pre-tested. If these did not occur, please provide the rationale for not doing so.

Response: This information is described in the original manuscript in the “Data Collection” paragraph in lines 286-289. The interview guides were pre-tested at the district hospital, War Memorial Hospital, in Navrongo with one mother and one midwife to assess the appropriateness of the questions and to determine the approximate time required to complete an interview. After this pre-test, we revised the interview guides based on the interviewers’ experiences and observations to improve the understandability of the questions. 

4) We note that you have indicated that data from this study are available upon request. PLOS only allows data to be available upon request if there are legal or ethical restrictions on sharing data publicly. For information on unacceptable data access restrictions, please see http://journals.plos.org/plosone/s/data-availability#loc-unacceptable-data-access-restrictions.

Response: The interview transcriptions cannot be shared publicly as it contains personally identifiable information and would therefore undermine the minimal risk ethical committee agreement and terms and conditions of consent. Interviews were confidential to enable freedom of expression, and participants consented to the study with the understanding that only anonymized quotations would be used in publications, not the entirety of the transcripts. Therefore, only illustrative excerpts from the transcripts, which qualify as the minimal data set, are included in the paper. Anonymized data can be made available to qualified researchers by request to the head of the data repository of the Navrongo Health Research Center: Peter Wontou (peter.wontuo@navrongo-hrc.org).

Comments from Reviewer 1:

1) Introduction section: Update literature on prevalence of HIV among pregnant women at ANC in study area. There is also relevant literature in Ghana e.g by Osei et al 2015 on prevention of MTCT of HIV and should be included. Probable reason for very low uptake of ARV services among HIV-positive pregnant women should be stated.

Response: Updated data on the prevalence of HIV among pregnant women in the study area at the national (lines 65-67), regional (lines 80-82), and district level (lines 82-84) was included. Findings from the Osei et al. 2016 study on PMTCT of HIV is described in the fifth paragraph (lines 85-93) to highlight the rationale of our study. The probable reasons for low ARV coverage among HIV-positive pregnant women was added (lines 72-74).

2) Study location: Provide more information on the health system context by way of facilities or services available at the different levels of health facilities especially as it relates to ANC and prevention of MTCT. Dates or period study was conducted should be stated.

Response: The structure of Ghana’s health care system as organized by different levels of health facilities as described in lines 101-106 of the “Study Location” section. The following paragraph (lines 107-117) provides information on the HIV-related/PMTCT services that are available at the various levels of facilities. The dates of the data collection phase were added in lines 467-468 under the “Data Collection” section.

3) Population and sampling approach: Consider systematically organizing this section into paragraphs of first the study population, sampling at the district level, zones and health facilities by levels of care for clarity. Please indicate the reasons for the sampling approaches used for the different populations.

Response: As suggested by the reviewer, we have restructured the “Study Population and Sampling Approach” section for clarity. The “Study Population” section (lines 248-251) describes the study population. The “Sampling Approach” section (lines 252-283) was reorganized to funnel from district zones, health facilities, and participants. Thank you for this insightful suggestion.

4) Results: Please provide demographic characteristics of study participants. Please include information on availability of treatment for the prevention of MTCT of HIV since it could also impact on compliance to the guidelines being assessed. Additionally it could be an important discussion point.

Response: A table of demographic characteristics of study participants was added to the Results section in lines 500-516. Excerpts from the interviews describing the availability and accessibility of antiretrovirals for the treatment of HIV at ANC clinics were included in the “Process of HIV Testing and Counseling” of the Results section in lines 598-613. 

5) Consenting: Reason provided for oral consenting is not accurate since most respondents were health workers and may be considered as a limitation instead.

Response: Thank you for pointing this out. We agree with this comment and have updated the statement about the reason for oral consent in the “Statement of Ethics” section (lines 496-497). We have amended the statement to read more appropriately.

---

## [Decision Letter · Decision Letter 1]

28 Feb 2022

PONE-D-20-35332R1Compliance to HIV testing and counseling guidelines at antenatal care clinics in the Kassena-Nankana districts of northern Ghana: A qualitative studyPLOS ONE

Dear Dr. Aborigo,

Thank you for submitting your manuscript to PLOS ONE. After careful consideration, we feel that it has merit but does not fully meet PLOS ONE’s publication criteria as it currently stands. Therefore, we invite you to submit a revised version of the manuscript that addresses the points raised during the review process. Please respond carefully to all of the reviewer comments when revising your manuscript, incorporating the clarifications and presentational improvements they have requested.

We look forward to receiving your revised manuscript.

Kind regards,

Jamie Males

Staff Editor

PLOS ONE

Journal Requirements:

Reviewers' comments:

Reviewer's Responses to Questions

**Comments to the Author**

1. If the authors have adequately addressed your comments raised in a previous round of review and you feel that this manuscript is now acceptable for publication, you may indicate that here to bypass the “Comments to the Author” section, enter your conflict of interest statement in the “Confidential to Editor” section, and submit your "Accept" recommendation.

Reviewer #2: (No Response)

2. Is the manuscript technically sound, and do the data support the conclusions?

Reviewer #2: Partly

3. Has the statistical analysis been performed appropriately and rigorously? 

Reviewer #2: N/A

4. Have the authors made all data underlying the findings in their manuscript fully available?

Reviewer #2: Yes

5. Is the manuscript presented in an intelligible fashion and written in standard English?

Reviewer #2: No

6. Review Comments to the Author

Reviewer #2: First, I should say, I have enjoyed reading this piece of work. Understanding adherence to a given guideline in a clinical setting is an important area of research that potentially may inform the effort to enhance the quality of HIV testing services in Ghana in particular and SSA in general. I only have a few comments to enhance the manuscript's readability and technical soundness.

Introduction

Some of the paragraphs need a revisit. For example, the paragraph from 55-63 is hard to follow, given that it began with a very long sentence. It also specifies "The third and fourth components of this strategy . . .", but not sure if the reader will pick up which third & fourth components the authors are referring to.

It would be more informative in lines 65-67 if the authors provided the period and the change in ANC service use proportion.

Methods

The methods section provides a helpful and detailed description of how the study was conducted to address the research question. A few things to have a look at:

The authors have indicated in Line 149-150 that Grounded Theory underpinned the study design. There are various schools of thought on Grounded Theory as methodology and method one can apply in qualitative studies. Please provide more information on which (whether the classical/traditional or constructive grounded theory informed the inquiry process – subsequently reflect how the framework informed your process of data collection, sampling, & analysis. Please also include references in your method section as appropriate.

In Line 160-163, the authors described as they have used theoretical sampling, which allows them to set criteria based on the research question, and ongoing analysis to recruit more participants from various population groups and settings. They have also said they used random sampling to document ethnic variation of any. This looks a purposive sampling as a random sample is not commonly used term in a qualitative study.

As the authors have provided background to describe the health care system in Uganda, several health facilities provide ANC services with midwives. So, in Lines 144-165, it would help your readers know the types and how many health facilities were selected. Also, please explain why you selected those health facilities. This is crucial to the research objective provided in the paper, which aims to provide crucial information on whether there is adherence to the guide inline of HIV testing across different health facilities.

Data analysis and Results

The result section presented the findings using a pre-defined sub-heading/themes that are likely driven from the interview guides. As the authors have used open, axial, and selective coding underpinned by grounded theory, it would be more informative to present here the key themes that were selected/emerged from the account of participants than using themes of the interview guides. For instance, “Lack of privacy and breach of confidentiality” could be one example theme that the authors discussed but presented under the process of HIS Testing and counseling at ANC.

· In Line 215, it says the transcripts were corrected to facilitate coding. Not sure what the authors mean by that. Changing participants' wording while doing a verbatim transcription will risk the principle of authenticity, but there are situations where making the change are reasonable. It could be to maintain the grammatical flow of sentences as sometimes participants may provide an incomplete speech. Or to make an oral speech into written sentences and paragraphs. This has to be reflected in the quotes used in the result section. Some of the quotes are hard to follow. Line 326-330 is a good example of a quote with many pronouns, which makes it hard to follow the message. This is a typical example that transcribing correction/changing or adding additional information within [ ] required to help readers understand what the participant is referring to when they are using pronouns. Please revisit the quotes to ensure its readability.

· Involvement of male partners in either using the services or women's decision making to use HIV testing services seems overlooked in the method (Line 128-146), result (line 334-335 & 340 – "male unfriendly perception among healthcare providers"), and discussion section (line 552). Please provide context if the guideline also addresses how male partners should be involved in PMTCT & how the services providers should engage with them.

· There are a few grammatical issues (e.g., Line 280), and a few hard-to-follow sentences, especially the quotes (lines 343-344).

Discussion

If key themes emerged/were selected from the open, axial, and selective coding process, readers would expect a summary paragraph at the beginning of the discussion section. Since the results are presented differently, the discussion started with one of the key themes – setting HTC as a compulsory vs. voluntary option. Please reconsider revising the paragraph to summarize the attributes/themes that show loose adherence to the national guidelines on HTC in ANC settings.

Limitation:

The study to assess whether healthcare provides comply with the national guidelines on HIV testing used snowball sampling (Line 172-174) of women who used the services may have created bias. There is a chance that the healthcare providers are likely to suggest recruiting women whom they think can speak the best experiences they had in the health facilities when they receive HIV testing. As such, this limitation has to be reflected in the limitation of the study. Also, the study used interviews to assess compliance of a practice. A study design like ethnography would be more stronger to provide the existing practice of HIV testing. This can be reflected in the limitation as well.

Conclusion

In Line 604-615, strong recommendations have been made. However, based on this single study, the reader may not know if there are ongoing efforts related to the actions listed within the recommendations made by the authors. It is also not clear whether these actions recommended by the authors can be taken as effective measures to address the grab identified regarding healthcare providers' compliance in providing HIV testing services. Instead, it would be more pragmatic to suggest that more studies be conducted using different research designs (such as ethnography or observational quantitative designs) to directly measure healthcare providers' compliance in providing HIV testing and counseling services against the national guideline.

7. PLOS authors have the option to publish the peer review history of their article (what does this mean?). If published, this will include your full peer review and any attached files.

Reviewer #2: No

---

## [Author Response · Author response to Decision Letter 1]

12 Apr 2022

Manuscript Number: PONE-D-20-35332

Response to Reviewers

Comments from Reviewer 1:

First, I should say, I have enjoyed reading this piece of work. Understanding adherence to a given guideline in a clinical setting is an important area of research that potentially may inform the effort to enhance the quality of HIV testing services in Ghana in particular and SSA in general. I only have a few comments to enhance the manuscript's readability and technical soundness.

Introduction

Some of the paragraphs need a revisit. For example, the paragraph from 55-63 is hard to follow, given that it began with a very long sentence. It also specifies "The third and fourth components of this strategy . . .", but not sure if the reader will pick up which third & fourth components the authors are referring to.

Response: We reformatted this sentence into a list for clarity (lines 56-62).

It would be more informative in lines 65-67 if the authors provided the period and the change in ANC service use proportion.

Response: We provided quantitative data from prior research to illustrate the increase in ANC use from 2006 to 2017-2018 (lines 68-69).

Methods

The methods section provides a helpful and detailed description of how the study was conducted to address the research question. A few things to have a look at:

The authors have indicated in Line 149-150 that Grounded Theory underpinned the study design. There are various schools of thought on Grounded Theory as methodology and method one can apply in qualitative studies. Please provide more information on which (whether the classical/traditional or constructive grounded theory informed the inquiry process – subsequently reflect how the framework informed your process of data collection, sampling, & analysis. Please also include references in your method section as appropriate.

Response: We used the Straussian grounded theory. We utilized the fundamental characteristics of this approach, namely theoretical sampling, thematic saturation, constant comparative analysis, and memoing to guide the sampling approach, data collection, and data analysis, which is further discussed in the Data Collection section (lines 247-253). 

In Line 160-163, the authors described as they have used theoretical sampling, which allows them to set criteria based on the research question, and ongoing analysis to recruit more participants from various population groups and settings. They have also said they used random sampling to document ethnic variation of any. This looks a purposive sampling as a random sample is not commonly used term in a qualitative study.

Response: In the Kassena-Nankana districts, the Kassena ethnic group predominantly lives in the North and West zones while the Nankani ethnic group predominantly lives in the East and South zones. To ensure our study population included both ethnic groups, we randomly selected a zone to represent the Kassenas and a zone to represent the Nankanis. When selecting the health facilities, we used purposive sampling to select those that had a midwife providing antenatal care. 

As the authors have provided background to describe the health care system in Uganda, several health facilities provide ANC services with midwives. So, in Lines 144-165, it would help your readers know the types and how many health facilities were selected. Also, please explain why you selected those health facilities. This is crucial to the research objective provided in the paper, which aims to provide crucial information on whether there is adherence to the guide inline of HIV testing across different health facilities.

Response: We specified the number and types of health facilities in our sample (lines 203-204). 

Data analysis and Results

The result section presented the findings using a pre-defined sub-heading/themes that are likely driven from the interview guides. As the authors have used open, axial, and selective coding underpinned by grounded theory, it would be more informative to present here the key themes that were selected/emerged from the account of participants than using themes of the interview guides. For instance, “Lack of privacy and breach of confidentiality” could be one example theme that the authors discussed but presented under the process of HIS Testing and counseling at ANC.

Response: We organized the result section into the broader themes of the process of HTC, information provided during HIV counseling, voluntariness of HTC, and challenges with HTC to provide a general overview of the way in which HTC is provided. We agree that the theme of “Lack of Privacy and Breach of Confidentiality” is one that can stand alone, so we have made it a separate section (lines 532-544).

In Line 215, it says the transcripts were corrected to facilitate coding. Not sure what the authors mean by that. Changing participants' wording while doing a verbatim transcription will risk the principle of authenticity, but there are situations where making the change are reasonable. It could be to maintain the grammatical flow of sentences as sometimes participants may provide an incomplete speech. Or to make an oral speech into written sentences and paragraphs. This has to be reflected in the quotes used in the result section. Some of the quotes are hard to follow. Line 326-330 is a good example of a quote with many pronouns, which makes it hard to follow the message. This is a typical example that transcribing correction/changing or adding additional information within [ ] required to help readers understand what the participant is referring to when they are using pronouns. Please revisit the quotes to ensure its readability.

Response: The transcripts were edited to maintain grammatical flow and proper syntax of sentences as some of the interviews were conducted in the local languages and the transcribers had little time to review the grammar and syntax of their transcriptions. The edits were minimal, and care was taken to preserve the meaning of the participants’ responses. We have also edited the quotes to improve comprehension.

Involvement of male partners in either using the services or women's decision making to use HIV testing services seems overlooked in the method (Line 128-146), result (line 334-335 & 340 – "male unfriendly perception among healthcare providers"), and discussion section (line 552). Please provide context if the guideline also addresses how male partners should be involved in PMTCT & how the services providers should engage with them.

Response: The national guidelines on PMTCT indicate that HIV testing and counseling services should be encouraged and offered to male partners as well but does not provide further information on how service providers should engage with them. This has been incorporated in the Methods section (lines 179-180).

There are a few grammatical issues (e.g., Line 280), and a few hard-to-follow sentences, especially the quotes (lines 343-344).

Response: We have edited the grammar in Line 280 (now line 314) and the quote in Lines 343-344 (now lines 413-414) to improve readability.

Discussion

If key themes emerged/were selected from the open, axial, and selective coding process, readers would expect a summary paragraph at the beginning of the discussion section. Since the results are presented differently, the discussion started with one of the key themes – setting HTC as a compulsory vs. voluntary option. Please reconsider revising the paragraph to summarize the attributes/themes that show loose adherence to the national guidelines on HTC in ANC settings.

Response: We revised the first paragraph of the discussion to summarize the attributes/themes that show loose adherence to the national guidelines on HIV testing and counseling in antenatal care settings (lines 622-629).

Limitation

The study to assess whether healthcare provides comply with the national guidelines on HIV testing used snowball sampling (Line 172-174) of women who used the services may have created bias. There is a chance that the healthcare providers are likely to suggest recruiting women whom they think can speak the best experiences they had in the health facilities when they receive HIV testing. As such, this limitation has to be reflected in the limitation of the study. Also, the study used interviews to assess compliance of a practice. A study design like ethnography would be more stronger to provide the existing practice of HIV testing. This can be reflected in the limitation as well.

Response: We agree with the reviewer’s additional suggested limitations and have edited the limitation section to reflect this (lines 717-730).

Conclusion

In Line 604-615, strong recommendations have been made. However, based on this single study, the reader may not know if there are ongoing efforts related to the actions listed within the recommendations made by the authors. It is also not clear whether these actions recommended by the authors can be taken as effective measures to address the grab identified regarding healthcare providers' compliance in providing HIV testing services. Instead, it would be more pragmatic to suggest that more studies be conducted using different research designs (such as ethnography or observational quantitative designs) to directly measure healthcare providers' compliance in providing HIV testing and counseling services against the national guideline.

Response: The recommendations were made to offer potential interventions and strategies to improve standardization and quality of HIV testing and counseling services. We agree with the reviewer that further research, such as ethnographies and observational studies, should be conducted to capture actual practices of HIV testing and counseling and have revised the conclusion to reflect this (lines 757-759).

---

## [Editor Report · Decision Letter 2]

7 Sep 2022

Compliance to HIV testing and counseling guidelines at antenatal care clinics in the Kassena-Nankana districts of northern Ghana: A qualitative study

PONE-D-20-35332R2

Dear Dr. Aborigo,

We’re pleased to inform you that your manuscript has been judged scientifically suitable for publication and will be formally accepted for publication once it meets all outstanding technical requirements.

Kind regards,

James Mockridge

Staff Editor

PLOS ONE

---

## [Editor Report · Acceptance letter]

21 Sep 2022

PONE-D-20-35332R2 

Compliance to HIV testing and counseling guidelines at antenatal care clinics in the Kassena-Nankana districts of northern Ghana: A qualitative study 

Dear Dr. Aborigo:

I'm pleased to inform you that your manuscript has been deemed suitable for publication in PLOS ONE. Congratulations! Your manuscript is now with our production department. 

Kind regards, 

on behalf of

Dr James Mockridge 

Staff Editor

PLOS ONE